

# Ananke: temporal clustering reveals ecological dynamics of microbial communities

Michael W. Hall[1], Robin R. Rohwer[2], Jonathan Perrie[3], Katherine D. McMahon[4,5] and Robert G. Beiko[3]

[1] Faculty of Graduate Studies, Dalhousie University, Halifax, Nova Scotia, Canada
[2] Environmental Chemistry and Technology Program, University of Wisconsin-Madison, Madison, WI, United States of America
[3] Faculty of Computer Science, Dalhousie University, Halifax, Nova Scotia, Canada
[4] Department of Civil and Environmental Engineering, University of Wisconsin-Madison, Madison, WI, United States of America
[5] Department of Bacteriology, University of Wisconsin-Madison, Madison, WI, United States of America

Corresponding authors
Michael W. Hall, mike.hall@dal.ca
Robert G. Beiko, beiko@cs.dal.ca

## ABSTRACT

Taxonomic markers such as the 16S ribosomal RNA gene are widely used in microbial community analysis. A common first step in marker-gene analysis is grouping genes into clusters to reduce data sets to a more manageable size and potentially mitigate the effects of sequencing error. Instead of clustering based on sequence identity, marker-gene data sets collected over time can be clustered based on temporal correlation to reveal ecologically meaningful associations. We present Ananke, a free and open-source algorithm and software package that complements existing sequence-identity-based clustering approaches by clustering marker-gene data based on time-series profiles and provides interactive visualization of clusters, including highlighting of internal OTU inconsistencies. Ananke is able to cluster distinct temporal patterns from simulations of multiple ecological patterns, such as periodic seasonal dynamics and organism appearances/disappearances. We apply our algorithm to two longitudinal marker gene data sets: faecal communities from the human gut of an individual sampled over one year, and communities from a freshwater lake sampled over eleven years. Within the gut, the segregation of the bacterial community around a food-poisoning event was immediately clear. In the freshwater lake, we found that high sequence identity between marker genes does not guarantee similar temporal dynamics, and Ananke time-series clusters revealed patterns obscured by clustering based on sequence identity or taxonomy. Ananke is free and open-source software available at https://github.com/beiko-lab/ananke.

## INTRODUCTION

Phylogenetic marker-gene sequencing has revolutionized our understanding of microbial ecology. Nearly every conceivable habitat has been profiled using markers such as the 16S ribosomal RNA (rRNA) gene. These studies have revealed a hitherto unappreciated degree of diversity among both well-studied and novel microorganisms (*Lynch & Neufeld,*

*2015*). A single sample provides a detailed view of a microbial community at one given point in time, but time-series sampling is increasingly used to track changes in a microbial community, often in connection with changes in the environment. Examples of time-series sampling include the tracking of microbial succession in the gut of a developing infant (*Koenig et al., 2011*), demonstrating the existence of a "microbial seed bank" in a marine environment (*Caporaso et al., 2012*), and showing differences in temporal variability of human oral, gut, and skin microbial communities across individuals (*Flores et al., 2014*).

The large amount of data generated in microbial marker-gene surveys can present a significant impediment to analysis; a single data set can contain millions of unique sequences, including real variants and products of sequencing error. Clustering methods are often used to reduce the magnitude of the data and minimize the impact of sequencing errors. Traditionally, the most common clustering approach is to merge sequences into operational taxonomic units (OTUs) at a pre-defined sequence-identity threshold, often 97% (*Koenig et al., 2011*; *Caporaso et al., 2012*; *Flores et al., 2014*; *Shade et al., 2013*; *David et al., 2014*; *Caporaso et al., 2011*). Although sequence-identity-based OTU clustering can streamline and simplify analyses, it suffers from limitations. Sequences from ecologically distinct community members can be lumped together into the same OTU if their marker genes have high sequence identity, thus treating them as a single entity in spite of their ecological differences (*Tikhonov, Leach & Wingreen, 2015*; *Eren et al., 2013*). This can diminish the effectiveness of analyses that treat OTUs as homogeneous entities, such as co-occurrence network analysis (*Beiko, 2015*). The common sequence-identity threshold of 97% is also seen as a proxy for species boundary, but the high accuracy of modern sequencers (*Schirmer et al., 2015*) allows us to confidently investigate marker-gene data at a finer resolution. Several new methods, such as DADA2, oligotyping, and minimum entropy decomposition, have been developed to harness the accuracy of modern sequence data to increase the resolution of marker gene analyses (*Callahan et al., 2016*; *Mark Welch et al., 2014*; *Eren et al., 2015*). These tools indicate a shift away from 97% sequence-identity OTUs and toward more precise sequence variants.

Methods that construct clusters based on attributes more closely linked to ecological properties can overcome the limitations of sequence-identity-based OTUs while retaining the benefits of clustering. For example, distribution-based clustering has been used to split OTUs when the member sequences have distinct distributions across samples, minimizing inappropriate aggregation (*Preheim et al., 2013*). With time-series data, sequences can be clustered based on correlated changes in relative abundance, which emphasizes temporal cohesion at the possible expense of taxonomic coherence. This paper introduces Ananke, a new algorithm and software package that clusters sequences based on temporal dynamics rather than sequence identity. Ananke, the consort of Chronos in Greek mythology, is the deity representing compulsion and necessity. Ananke generates time-series clusters (TSCs) by grouping marker gene sequences based on consistent changes in their relative abundance over time. We describe Ananke's clustering algorithm, as well as its interactive tool for visualizing results. This paper demonstrates Ananke's high fidelity in detecting ecological patterns and events using simulated time-series data, and demonstrates Ananke's utility using two 16S rRNA gene time-series data sets. Ananke TSCs had defined ecological

roles in a human gut data set, reflected seasonal dynamics in a temperate lake data set, and identified subtle patterns in each that may represent previously undescribed ecological processes.

## MATERIALS AND METHODS

### Input data

Ananke requires only the sequence data and time points as input. The sequence data can be any FASTA-formatted data, including but not limited to 16S rRNA gene amplicon sequences. Sequences can be preprocessed (quality filtered, trimmed, ambiguous nucleotides removed, etc.) beforehand with users' preferred methods. The time point data is a metadata file that relates the sample names to their relative sampling time.

### Data tabulation and storage

Ananke tabulates the abundance of each unique sequence at each time point, resulting in an $m \times n$ time-series matrix where $m$ is the number of unique sequences and $n$ is the number of time points. To reduce space on disk and in memory, this data is stored in compressed sparse row format in an HDF5 file (*The HDF Group, 1997*). The flexible HDF5 format allows for storage of all necessary data and metadata in a single file using a binary representation. Taxonomic classifications and traditional sequence-identity-based OTUs can be computed with users' preferred pipelines and stored in the same HDF5 file. Since Ananke operates on unique sequences rather than sequence-identity-based OTUs, data filtering is a necessary step for larger data sets. Unique sequences can be filtered based on the abundance of the sequence or the proportion of samples in which they appear. Ananke can use raw sequences that are quality filtered with any pipeline. Using raw sequences can be a beneficial approach as it maximizes the available information by avoiding any unnecessary aggregation of sequences. This comes at the cost of larger data magnitude, which can be handled by increased computational power, or a sequence filtering step. It is also important to take caution in the interpretation of the TSCs that result from raw sequences as the uncorrected reads will contain sequencing errors, chimeras, and other artefacts. Ananke can import raw sequences from a FASTA file, or denoised sequences from DADA2 (*Callahan et al., 2016*). If importing from DADA2, we recommend increasing the OMEGA_A parameter to minimize overly aggressive aggregation. Denoising methods, such as DADA2, minimum entropy decomposition (*Eren et al., 2015*), or oligotyping (*Mark Welch et al., 2014*) are a useful complement to Ananke, as they use a conservative approach to correct sequences that reduces data set size but is not as destructive to the underlying patterns as traditional sequence-identity clustering. Denoised sequences generated by any method can be imported into Ananke if they are first converted to FASTA format. If the user chooses to denoise as a pre-processing step to Ananke, we recommend using conservative correction parameters to avoid inadvertently degrading temporal patterns, especially those resulting from rare sequences.

### Calculating distance between time series

Ananke uses the short time-series (STS) distance (*Möller-Levet et al., 2003*) to compute the distances between each pair of unique sequences at each time point. This distance

represents the degree of dissimilarity between the sequences' temporal profiles. Before computing the STS distance, the sequence counts for each time point are normalized by dividing by each time point's sequence depth. Then each sequence's temporal profile, $x_i$, is standardized to $Z$-scores as in *Möller-Levet et al. (2003)*:

$$z_i = \frac{x_i - \bar{x}_i}{s_{x_i}}$$

where $\bar{x}_i$ is the mean and $s_{x_i}$ is the standard deviation of the $i$th sequence's temporal profile. The squared distance between two standardized temporal profiles, $z_i$ and $z_j$, is computed using the formula:

$$d_{STS}^2 = \sum_{k=0}^{n-1} \left( \frac{z_{i,k+1} - z_{i,k}}{t_{k+1} - t_k} - \frac{z_{j,k+1} - z_{j,k}}{t_{k+1} - t_k} \right)^2$$

where $i$ and $j$ index the $m$ unique sequences, and $k$ indexes the $n$ time points. For each unique sequence there are $n-1$ slopes between the $n$ consecutive time points. For a given pair of unique sequences, the differences between their slopes are squared and summed to obtain their STS distance. The STS distances are divided by the maximum distance in the data set so that they fall in the closed interval $[0, 1]$. The STS distance is the recommended measure as it takes the sample order (i.e., temporal gradient) into account; however, other measures such as Euclidean and Manhattan distances are available to use. To control for data compositionality within each sample, users can select an optional centered log ratio transform with count zero multiplicative zero imputation before distance calculations, using the methods from the CoDaSeq R library (*Gloor et al., 2016*).

## Clustering of time-series distances

The unique sequence pairwise STS distance matrix is clustered into Ananke TSCs by the DBSCAN algorithm (*Ester et al., 1996*) implemented in the scikit-learn Python library (*Pedregosa et al., 2011*). This algorithm requires two parameters: `min_samples`, and $\epsilon$. The `min_samples` parameter is set to 2 to prevent singletons from forming their own Ananke TSCs, and instead places them into the "noise bin" which contains all unclustered singleton sequences. These "noise" sequences are those that share no similar temporal dynamics, at a given $\epsilon$ value, with any other sequence, and as such both rare and highly abundant sequences can be labeled as noise. Additionally, rare sequences that appear in only one time point, if not filtered out by an abundance filter, will form TSCs with all other sequences that appear only at that time point. As $\epsilon$ is increased, fewer sequences will be labeled as noise, but some TSCs will grow too large in size to be useful. Ananke allows for interactive exploration of the parameter space by pre-computing results over a range of $\epsilon$ values. Run times and memory usage for the various steps in the Ananke computational pipeline are given in Table 1.

## Visualization of time-series clusters

The Ananke-UI facilitates data exploration with an interactive application built with Shiny (*Chang et al., 2015*), a library for the R programming language (*R Core Team, 2015*). Ananke-UI imports the results file and plots the temporal profiles of Ananke TSCs, allowing

**Table 1   Run time (in seconds) and memory usage (in MB) of the steps in the Ananke pipeline.**

| Step | Stool, denoised | | Lake, full | |
| --- | --- | --- | --- | --- |
| | Time (s) | Mem. (MB) | Time (s) | Mem. (MB) |
| Tabulate/Import | 3.8 | 126.6 | 609.1 | 2189.9 |
| Filter | – | – | 833.6 | 316.5 |
| Cluster | 36.3 | 719.7 | 733.1 | 10531.6 |
| Add Classifications | 2.3 | 106.9 | 11.1 | 221.6 |
| Add Clusters | 1.8 | 115.2 | 6.1 | 123.2 |
| Total (time) | 44.2 | – | 2193.0 | – |
| Max (memory) | – | 719.7 | – | 10531.6 |

users to interactively explore the effects of the clustering parameter $\epsilon$ in the browser-based application. We recommend that users begin exploring at the $\epsilon$ that provides the largest number of TSCs, and therefore the greatest separation of sequences. The user interface presents the taxonomic classifications and sequence-identity-based OTU assignments for each unique sequence in an Ananke TSC, allowing users to compare different clustering methods. The interface allows the user to explore their sequence identity-based clusters (e.g., traditional 97% OTUs) with the constituent unique sequences coloured by their time-series cluster membership. This allows temporal inconsistencies within an OTU to be identified at a glance.

## Generation of simulated data

Ecological patterns were simulated to provide a test set with known ground-truth cluster assignments. We simulated six types of temporal patterns: extinction, arrival, seasonality, conditional rarity (*Shade & Gilbert, 2015*), and stationary with low and high variance (Fig. 1). A template relative abundance profile was generated for each pattern and 100 random trials based on each template were created by adding additional random noise and scaling by a random factor. The simulations were repeated for different time-series lengths (25, 100, 250, 500, and 1,000 time points). The simulated temporal profiles were clustered over a range of $\epsilon$ clustering parameter values, and the adjusted mutual information (AMI) score (*Vinh, Epps & Bailey, 2010*) with respect to the ground-truth was used as a measure of cluster quality. The AMI score is a chance-corrected version of the mutual information score that accounts for the amount of agreement between two sets of clusters that is expected to be due to chance. It has been shown to be a better indication of cluster quality than mutual information or normalized mutual information scores (*Vinh, Epps & Bailey, 2010*). The highest achieved AMI across the computed $\epsilon$ parameters was reported. The code to generate simulations is available in the Ananke software package through the `simulation` and `score_simulation` subcommands.

## Human-associated and environmental data

Two biological time-series data sets were analyzed using Ananke. From *David et al. (2014)*, we analyzed the 191 faecal samples of "Subject B" taken on a nearly daily basis for a year. These data were retrieved from the European Bioinformatics Institute (EBI) under

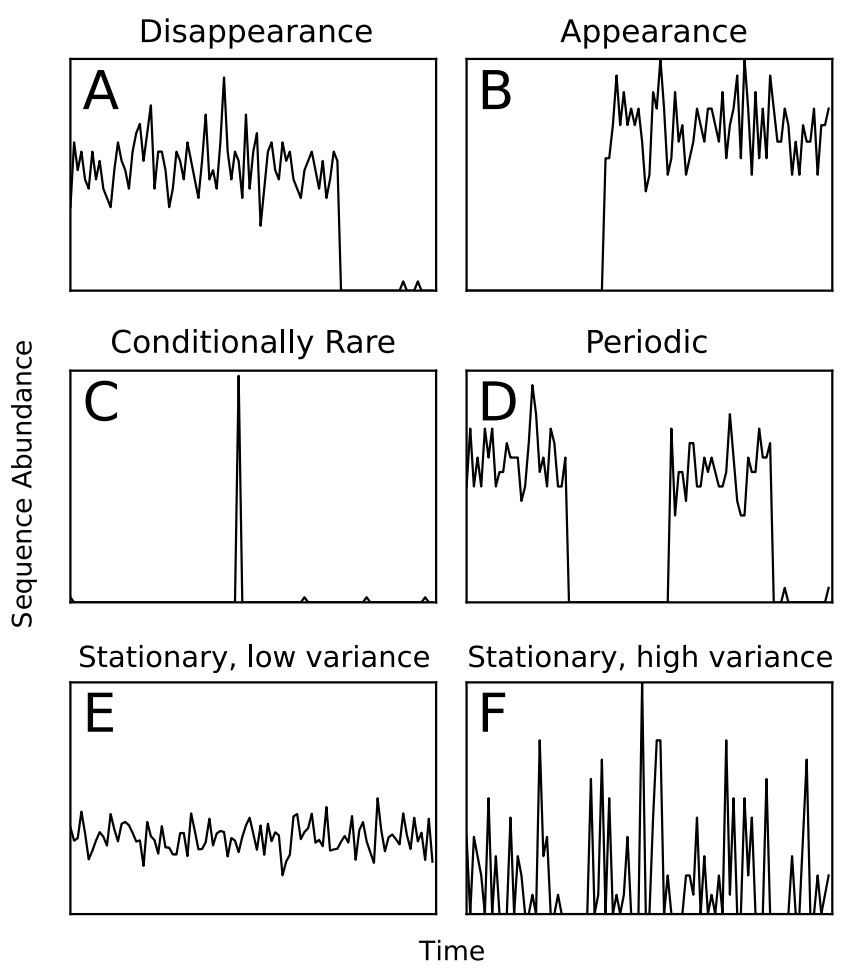

**Figure 1** Examples of the six types of simulated temporal patterns.

 The second data set is comprised of 96 time points from an eleven-year time series of Lake Mendota in Wisconsin, USA. Sequences and metadata were retrieved through EBI under accession PRJEB14911. For both data sets, Ananke TSCs were computed over a parameter range of $\epsilon = 0.01$ to $\epsilon = 1$ with a step size of 0.01. For comparison with sequence-identity clustering, sequences were clustered into 97% OTUs using the UPARSE pipeline (*Edgar, 2013*) at 97% identity. For the faecal data, all unique denoised sequences were classified with the Ribosomal Database Project naïve Bayesian classifier v2.2 (RDP classifier) at a minimum 60% posterior probability (*Wang et al., 2007*) trained against GreenGenes revision 13_8 (*McDonald et al., 2012*) via QIIME v1.9.0 (*Caporaso et al., 2010*). For the lake data, unique sequences with greater than 98% sequence identity to references in the Freshwater Training set (FreshTrain) (*Newton et al., 2011*) were classified with the RDP classifier at a minimum 80% posterior probability trained against the FreshTrain, and the remaining unique sequences were classified with the RDP classifier at a minimum 70% posterior probability trained against GreenGenes revision 13_8 via the TaxAss workflow (http://www.github.com/McMahonLab/TaxAss).

## Availability of software and data

The Ananke software, which includes the Python-based clustering algorithm, the R- and Shiny-based visualization platform, and associated documentation, is available on GitHub (http://github.com/beiko-lab/ananke and http://github.com/beiko-lab/ananke-ui). Scripts for reproducing the analyses, including data retrieval, sequence-identity clustering, taxonomic classification, and the Ananke pipeline are available at https://github.com/mwhall/Ananke_PeerJ. Ananke HDF5 data files for the lake and stool data sets are available on figshare (doi: 10.6084/m9.figshare.c.3707938.v1).

# RESULTS AND DISCUSSION

## Building clusters with Ananke

The goal of Ananke is to group unique marker-gene sequences that are "dynamically similar" (i.e., that correlate strongly over time) into clusters (*Tikhonov, Leach & Wingreen, 2015*). This general approach has been used to bin metagenomic sequences for the purpose of genome assembly (*Sharon et al., 2013*), whereas our method focuses on single genes that are used to track phylogenetically distinct groups. Briefly, the clustering algorithm proceeds as follows: (1) sequences are dereplicated and the time series are tabulated for each unique sequence, (2) data are filtered to remove sequences with sparsely sampled time series, (3) the short time-series (STS) distance (*Möller-Levet et al., 2003*) is calculated between each pair of unique sequences, (4) the resulting distance matrix is clustered into Ananke time-series clusters (TSCs) with DBSCAN (*Ester et al., 1996*), and (5) the Ananke TSCs are visualized and presented alongside sequence metadata.

The STS distance measure was designed for sampling schemes that are uneven and contain relatively few time points (*Möller-Levet et al., 2003*). Unlike other measures such as the Euclidean distance that are commonly used for clustering, the order of samples is important for the STS distance. The DBSCAN clustering algorithm was chosen for several reasons. DBSCAN can define outlier points as noise and remove them, rather than creating spurious clusters or adding irrelevant sequences to a cluster. DBSCAN is also an efficient method both in terms of memory usage and run time. DBSCAN requires a neighbourhood size clustering parameter, denoted by $\epsilon$, rather than a parameter that prespecifies the number of desired clusters, which other common clustering methods require. This is a more intuitive parameterization that is similar to sequence-identity clustering, as $\epsilon$ controls the granularity of the clusters. A smaller $\epsilon$ value implies clusters of sequences with more similar temporal profiles, whereas a larger $\epsilon$ would combine sequences with more disparate patterns.

## Assessing accuracy of Ananke with simulated data sets

Assessing cluster quality in a biological data set is a difficult task since no ground truth exists for comparison. To assess Ananke's cluster quality, we generated six artificial patterns of temporal variation that represent ecological events or patterns that users may wish to identify in a biological data set (Fig. 1). Appearance, disappearance, and conditional rarity (*Shade et al., 2014*) patterns may indicate responses to environmental changes, so it is important that Ananke clusters them appropriately. Periodic patterns often reflect

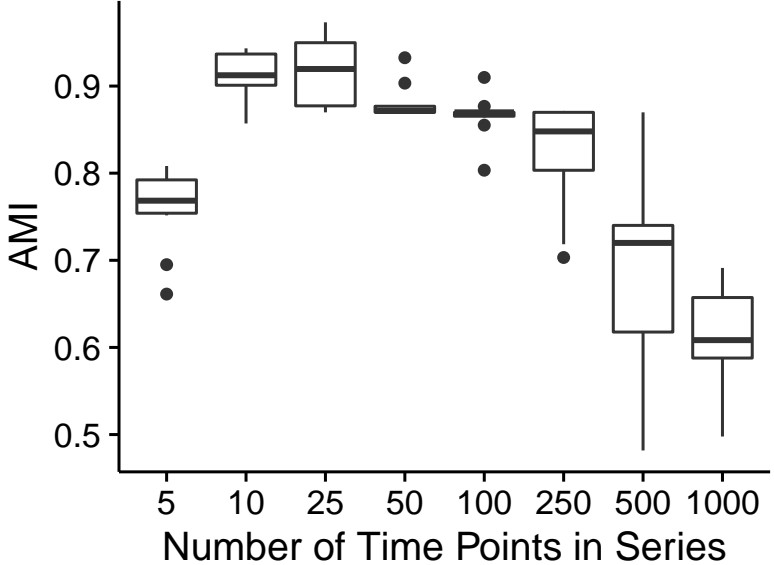

**Figure 2** **Simulated time-series clustering accuracy.** AMI scores for Ananke TSCs reconstructed from simulated time-series data sets of varying lengths. Boxplots of AMI scores across 10 independent simulations are shown, with 100 sequences per simulated cluster.

seasonal changes in natural environments, so Ananke must cluster time-series profiles with coordinated increases and decreases over time. Patterns that are stationary with low variance represent organisms with consistent abundance over time, while patterns that are stationary with a high variance may also represent noisy or undersampled data. Templates of each time-series pattern were created, and the simulated data sets were generated by adding random noise and scaling to the templates. We used AMI (*Vinh, Epps & Bailey, 2010*) to quantify the agreement between the Ananke TSCs computed for the simulated profiles and the ground-truth patterns from which they were generated. The AMI scores provide a quantitative measure of the quality of Ananke TSCs, where a higher AMI reflects higher agreement with the ground-truth patterns.

Ananke yielded average AMI scores >0.8 on simulated time-series data sets with as few as ten time points (Fig. 2). However, AMI scores were considerably lower for time-series data sets with 500 (median AMI = 0.69) and 1,000 (median AMI = 0.62) time points. The drop in AMI scores for very long time-series data sets is a consequence of the STS distance metric. The sum of small differences, which are a result of random noise added to each point, can overwhelm the effect of the true pattern over a large number of time points. To reduce the impact of random noise, very long time series could be smoothed by averaging over a sliding window. This would reduce the magnitude of the slopes that are due to random noise, resulting in a smaller cumulative impact on the distance measure.

The majority of the simulations flagged low-variance and high-variance stationary time-series profiles as noise, or placed these two patterns into the same TSC, which prevented Ananke from achieving higher AMI scores. Ananke's algorithm has trouble clustering stationary time-series profiles because they lack large slopes to influence the STS distance measure. The STS distance measure does not provide enough information

to separate the low-variance from the high-variance stationary temporal patterns since there are no consistently present large slopes. Ananke's current focus is on the detection of distinct ecological patterns such as appearance, disappearance, and conditional rarity, but future incorporation of the overall variance of temporal profiles in addition to shared slope would allow Ananke to also focus on stationary profiles.

## Time-series clustering reveals temporal segregation of taxa in the human gut

We used the time-series data set from *David et al. (2014)* to demonstrate our method with human-associated samples. The data are 16S rRNA gene fragments from faecal samples taken at 191 time points over 318 days. There were 26,250,105 total sequences and 1,200,847 unique sequences. As a pre-processing step, these sequences were denoised with DADA2 (*Callahan et al., 2016*) using an OMEGA_A parameter of 1E−2, and default parameters were used otherwise. After denoising, one time point had a sequence count lower than 1,000 and was excluded, leaving 190 time points. DADA2 reduced the total data by 14% to 22,468,163 sequences and the unique sequences by 99% to 2,618 sequences. As the denoising step reduced the data magnitude significantly, no additional filtering step was required. After time-series clustering with Ananke, a maximum of 180 Ananke TSCs were found at $\epsilon = 0.1$, with an average Ananke TSC comprising 0.5% of the data set with 124,823 total sequences and 15 unique sequences (Fig. S1).

The sampled subject experienced food poisoning as a likely result of *Salmonella* sp. around day 159. The authors of the original study showed that the food-poisoning event divides the faecal microbial community into three clear segments from days 0–144, 145–162, and 163–240 (*David et al., 2014*). In Ananke TSCs this segregation is readily apparent (Fig. 3). Some Ananke TSCs disappear after the disturbance event, such as one containing *Lachnospiraceae* sequences (Fig. 3A), while others emerge in the environment after the illness, such as a second Ananke TSC containing sequences classified as *Lachnospiraceae* (Fig. 3C). During the food-poisoning disturbance, 108 conditionally rare unique sequences (accounting for 112,311 total sequences) demonstrated an increase in relative abundance and were assigned to the same Ananke TSC at $\epsilon = 0.16$ (Fig. 3B). The two most abundant sequences in this spike classify to *Enterobacteriaceae* (the family containing *Salmonella* sp.) and *Haemophilus parainfluenzae*. The remaining sequences belonged to various taxonomic groups including the genera *Leuconostoc*, *Weissella*, *Lactococcus*, and *Turicibacter* from the class *Bacilli*; *Clostridium* and *Veillonella* from the class *Clostridia*, including known pathogen *C. perfringens*; and several sequences from the genus *Acinetobacter*.

Ananke highlighted several smaller changes in the community in addition to the changes associated with the food-poisoning disturbance. Some of these changes, such as those shown in Fig. 4, occurred at a sub-OTU level. After day 22, two *Dorea* OTUs have their most abundant sequence decrease in relative abundance to below detection. A second sequence variant is introduced and persists until the food poisoning event around day 59, when it sees a decrease in relative abundance. In both cases, a third sequence variant appears. In OTU 6, classified to *Dorea*, this third variant does not persist and the second variant reappears. In OTU 505, classified to *Ruminococcus gnavus*, the third variant

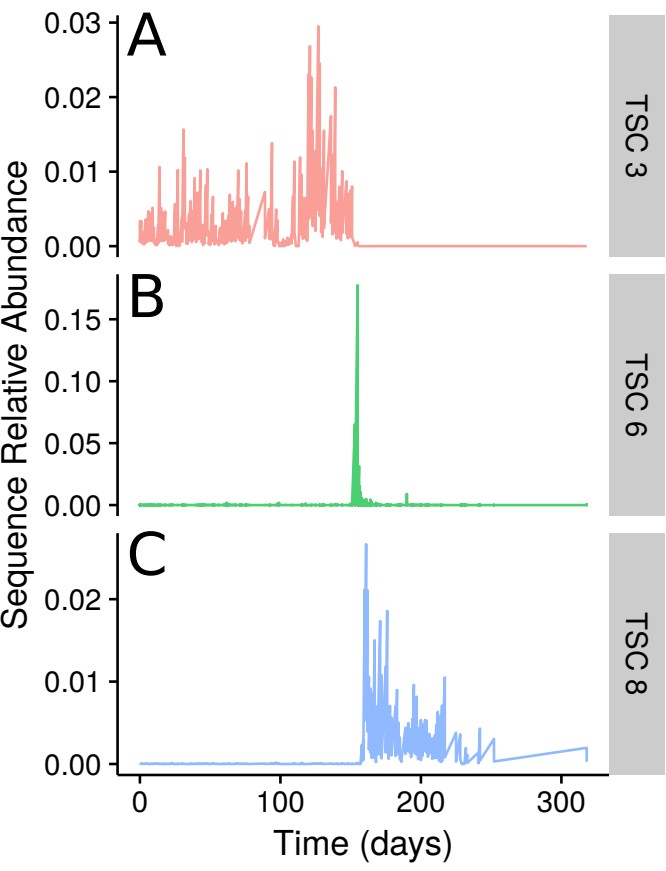

**Figure 3** **TSCs from a human faecal time series.** Examples of three select Ananke TSCs from human faecal 16S rRNA gene sequences that demonstrate the segmentation of the time series around a food poisoning event. (A) Two sequences classifying to *Lachnospiraceae* that are present before, but not after, the food poisoning event. (B) A total of 108 sequences that co-occur during the food poisoning event, including sequences classified to *Enterobacteriaceae*, *Haemophilus parainfluenzae*, and *Clostridium perfringens*. (C) Two additional *Lachnospiraceae* sequences that appear only after the food poisoning event.

persists, even alongside the second variant as it reappears. An OTU-based approach risks aggregating these sequence variants, thereby obscuring this subtle transition. Compared with OTU methods, Ananke provides an alternate, higher resolution method to highlight both clear and subtle partitioning of the profiles with respect to time.

## Seasonal dynamics in a freshwater lake are captured by time-series clustering

The second biological time-series data set is from Lake Mendota in Wisconsin, USA. This 16S rRNA gene amplicon data set spans eleven years with 96 total time points. There were 45,094,125 total and 3,058,132 unique sequences. Five time points had fewer than 1,000 sequences and were subsequently discarded, leaving 91 time points in the series. For Ananke clustering, the data were filtered to only include sequences with abundance $\geq 1,000$ counts, reducing the total data by 21% to 37,511,477 sequences and the unique sequences by 99% to 20,268 sequences. A maximum of 664 Ananke TSCs were found at $\epsilon = 0.09$, with an average TSC comprising 0.2% of the data set with 56,493 total sequences and 31

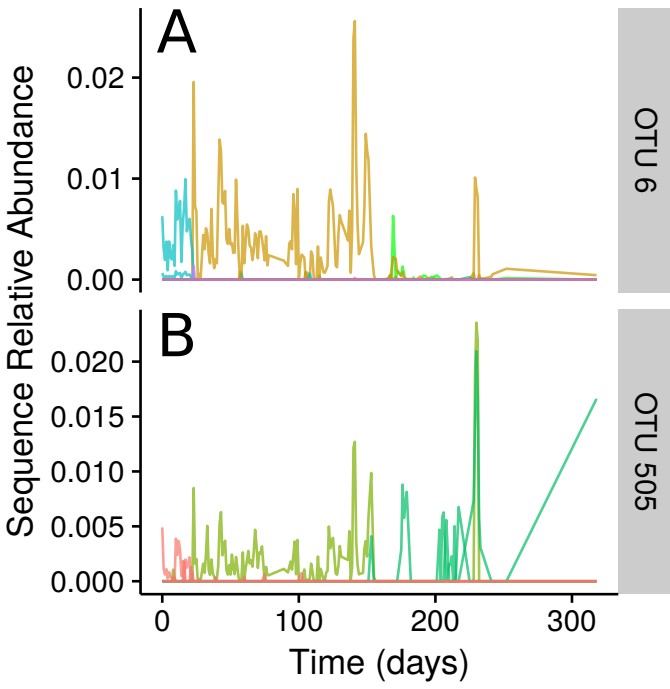

**Figure 4** **Inconsistencies within 97% sequence-identity OTUs.** (A) Plots of the time-series for the se-quences contained within OTU 6, with all sequences classifying to genus *Dorea*. (B) Time-series for the se-qeuences contained within OTU 505, with all sequences classifying to *Ruminococcus gnavus*. Sequences are coloured by their Ananke TSCs at $\epsilon = 0.2$, highlighting the fact that these sequences exhibit different dis-tributions across time despite being part of the same 97% sequence-identity OTU.

unique sequences (Figs. S2). This is in contrast to a recent analysis of this data set that grouped 97% OTUs from these sequences into only 14 clusters based on their annual peak (*Dam et al., 2016*). Ananke's clustering is based on the entire time series instead of a single temporal feature, which results in finer-resolution clusters.

In the Lake Mendota decade-long data set, Ananke identified seasonal patterns obscured in analyses using traditional 97% OTUs or taxonomy. Freshwater bacteria in this data set were named according to the freshwater training set (FreshTrain) nomenclature, where the taxa levels lineage, clade, and tribe approximate the Linnaean family, genus, and species (*Newton et al., 2011*). Ananke TSCs revealed both similarities between ranks of phylogenetically diverse organisms and fine-scale differences within taxa and OTUs.

The abundant freshwater *Bacteroidetes* lineage bacI is known to prefer high dissolved organic carbon, which often occurs during cyanobacterial or algal blooms (*Newton et al., 2011*). One of the most abundant bacI Ananke TSCs also included cyanobacterial reads from the common freshwater genus *Synechococcus* (Figs. 5A and 5B). The possibility of this type of co-occurrence is supported by a previous incubation study that found heterotrophic bacterial community composition correlates with the phytoplankton species (*Bagatini et al., 2014*). Ananke was able to identify this type of relationship in an observational time series, despite the fine-level taxonomy being unknown and the phylogenetic distance between the co-occurring groups. Discovering these correlations between distant organisms is

Peer**J**

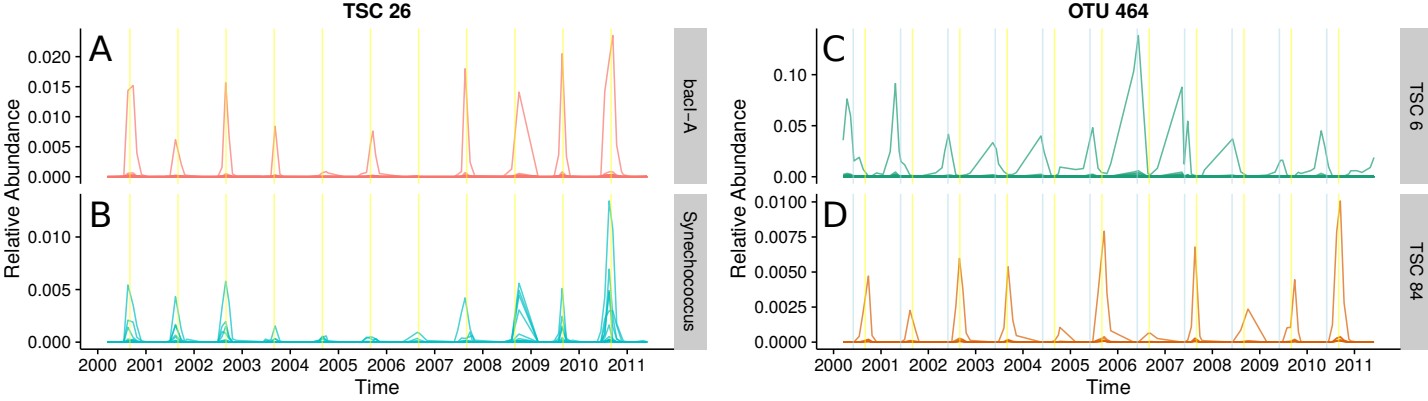

**Figure 5** **Seasonal patterns in a freshwater lake revealed by TSCs.** (A) and (B): Ananke TSCs can group sequences from distant taxonomic groups, highlighting shared temporal dynamics and suggesting possible associations. Ananke TSC 26 contains sequences classified to the heterotrophic *Bacteroidetes* bacI (A) and the cyanobacterial genus *Synechococcus* (B). TSC 26 displays periodicity with abundance peaking in September (yellow vertical line). Conversely, sequence-identity-based OTUs can contain sequences from multiple distinct TSCs. (C) and (D): Sequence-identity-based OTU 464, based on a 97% sequence-identity cut-off, contains sequences from the Luna1-A1 tribe that belong to two distinct TSCs, representing two distinct temporal patterns. Sequences from OTU 464 that belong to TSC 6 (C) peak around June (indicated by blue vertical lines), while sequences that belong to TSC 84 (D) peak around September (indicated by yellow vertical lines).

one advantage of Ananke over the distribution-based clustering by *Preheim et al. (2013)*, which identifies and removes inconsistencies within OTUs but is unable to group distantly related organisms that share distributional patterns. In many cases, it may be preferable to keep distant sequences from being placed into the same cluster, and for these cases distribution-based clustering is a more appropriate method. Taxonomically heterogeneous TSCs can be useful for generating hypotheses about potential interactions between members of a community, and Ananke's sequence identity-agnostic approach allows users to identify these potential relationships, which can then be validated with additional experimentation.

Ananke also identified ecological differences between closely related organisms. A single 97% OTU represented most of the actinobacterial Luna1-A1 tribe and erroneously included unclassified sequences that may belong to another closely related tribe. Two distinct Ananke TSCs reveal divergent ecological dynamics within this 97% OTU (Figs. 5C and 5D). The Luna1 lineage, which contains the tribe Luna1-A1 and three others, is one of the most abundant *Actinobacteria* in lakes and contains several candidate species including *Candidatus* "Aquiluna rubra" and *Candidatus* "Rhodoluna lacicola" (*Newton et al., 2011*; *Hahn, 2009*). Despite being considered the same taxonomic unit by traditional bioinformatic approaches, the two distinct temporal variants bloomed in June (Fig. 5C) and September (Fig. 5D). The sequences that peak in June classify to the tribe Luna1-A1. The sequences that peak in September only classify confidently down to the lineage level, but the gene fragments match most closely to tribe Luna1-A2. June and September are both months that represent transitions in the seasonal lake cycle: June is a period of rapid warming and switch from clear-water phase to cyanobacterial domination, while September is a period of cooling. Thus, it is reasonable to expect that taxonomically similar but ecologically distinct Luna1 members would be present in each month. The fine-scale

diversity revealed by Ananke can provide insights into the ecology of these tribes that would go unobserved in analyses even at the 97% OTU or lineage level.

The most dominant bacterial lineage in many freshwater lakes is the *Actinobacteria* acI. This lineage is made up of three major clades, acI-A, acI-B, and acI-C, which accounted for 10, 7, and 2% of all reads in the Lake Mendota data set, respectively. In Lake Mendota each of these three clades contained a single dominant sequence that accounted for 38, 75, and 65% of each clade's abundance. Since the ecology of these organisms is often studied at the clade level, the dynamics of these dominant sequences drive our understanding of the clades. Multiple Ananke TSCs were identified within each clade, many of which were both abundant and divergent from the dominant sequences (Fig. 6). All of the acI-C Ananke TSCs shared the autumn peak of the dominant acI-C sequence, but two Ananke TSCs accounting for 6% of all acI-C reads differed in terms of the duration of the peak or the relative intensities in different years. Four acI-A Ananke TSCs and one acI-B Ananke TSC displayed seasonal dynamics with peaks in May (indicated by blue vertical lines), some with a secondary peak in November (indicated by yellow vertical lines). These seasonal clusters account for 24 and 2% of each clade's abundance. These results indicate that the acI-A and acI-B clades may encompass more diverse life strategies than previously recognized. Additionally, many sequences in the divergent Ananke TSCs belong to unclassified tribes or to the broad ACK-M1 group, which indicates that the FreshTrain should be updated to include additional reference sequences. Ananke clustering was able to reveal these dynamics despite limits of the taxonomic reference, suggesting that Ananke could be especially insightful in other ecosystems where taxonomic analyses occur at even coarser levels because they lack a custom, curated reference database like the FreshTrain.

## Exploration of temporal clusters using Ananke-UI facilitates identification of potential microbial interactions

Unlike sequence-identity-based clustering where a static cut-off such as 97% sequence identity is used, there is no single $\epsilon$ parameter appropriate across multiple data sets. The choice of $\epsilon$ depends on properties such as the number of time points, diversity, and sequence depth of the data set. Users must explore Ananke's results and identify the $\epsilon$ parameter that best addresses their research questions. Decreasing $\epsilon$ results in Ananke TSCs containing sequences with more cohesive temporal profiles, while increasing $\epsilon$ assembles larger clusters containing sequences with more dissimilar temporal profiles (Fig. 7). Ananke and the associated user interface Ananke-UI allow users to visualize and explore Ananke TSCs and relevant metadata such as the taxonomic classification and sequence-identity-based OTU membership of an Ananke cluster's constituent unique sequences. Potential relationships between microorganisms can be uncovered using Ananke-UI by interactively exploring Ananke TSCs at various $\epsilon$ values. For example, two distinct Ananke TSCs in the lake data set were each taxonomically homogeneous with sequences from *Actinomycetales* or *Acidimicrobiales* at $\epsilon = 0.11$ (Figs. 7A–7B). When the $\epsilon$ value is increased to 0.12, these two Ananke TSCs merge into a single Ananke TSC (Fig. 7C). An overlay of constituent sequences' temporal profiles shows that both sets of sequences tend to increase and decrease in relative abundance cohesively, with the exception of one period where the

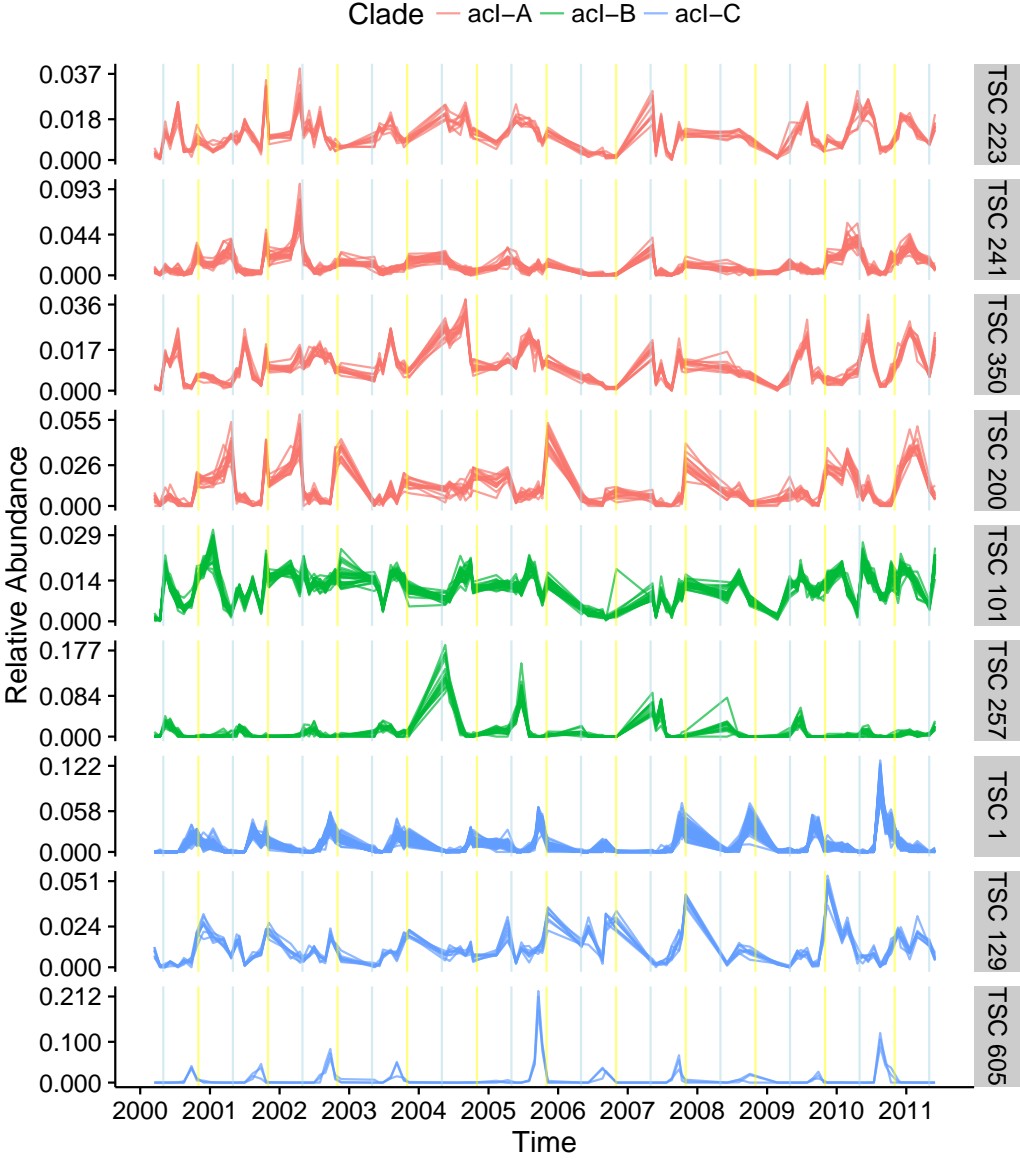

**Figure 6** **Temporal diversity within freshwater clades.** Distinct temporal dynamics can occur even within clades. The clades acI-A, -B, and -C comprise the abundant *Actinobacteria* lineage acI. Each clade contains one dominant temporal pattern (TSC 223, TSC 101, and TSC 1 for acI-A, -B, and -C, respectively), but Ananke identified additional clusters with divergent dynamics from the dominant sequences. Months in which population increases occur include May (indicated by blue vertical lines) and November (indicated by yellow vertical lines). Sequence abundances are normalized by dividing by sample sequence depth and then by sequence abundance in order to bring all sequences onto the same scale and highlight the time-series shapes.

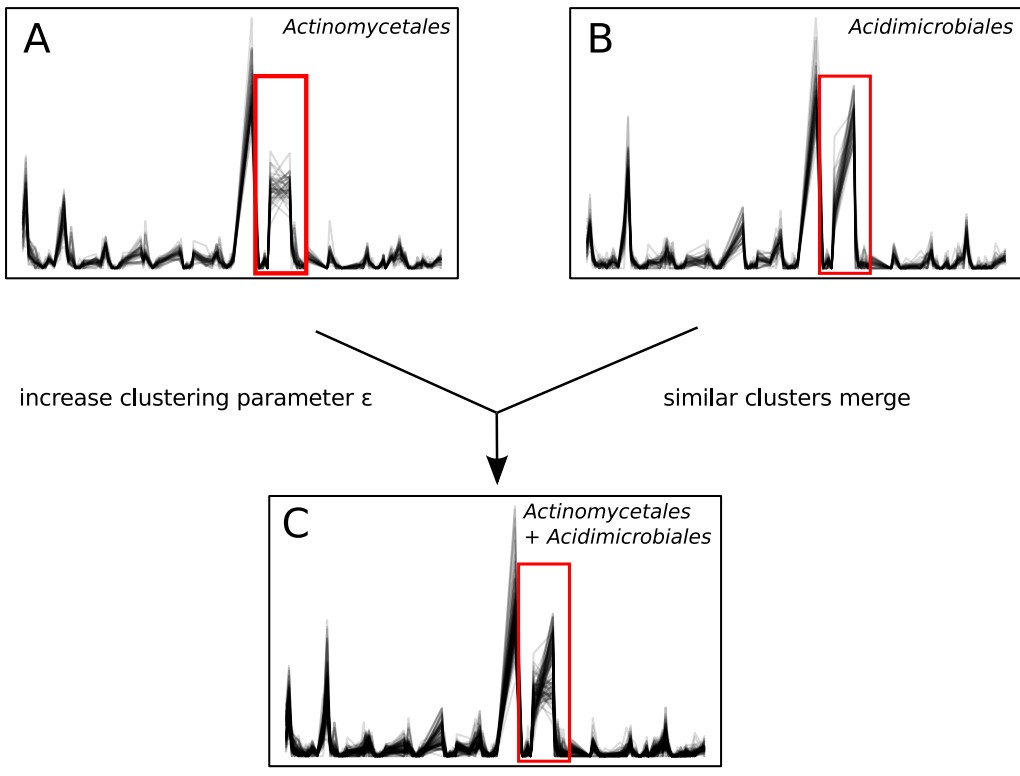

**Figure 7  Impact of the time-series clustering parameter $\epsilon$.** (A) and (B): two Ananke TSCs at clustering parameter $\epsilon = 0.11$. The cluster in (A) contains only sequences belonging to the order *Actinomycetales*, while (B) contains only sequences belonging to the order *Acidimicrobiales*. The red box highlights an area of the temporal profile that differs between the two TSCs. (C) When the clustering parameter is increased to $\epsilon = 0.12$, these two similar TSCs merge into a more taxonomically heterogeneous cluster.

two subclusters show divergent patterns of temporal abundance. By highlighting these temporal similarities, Ananke can aid in generating hypotheses about the relationships between microorganisms in a comparable way to other techniques like co-occurrence networks. Conversely, temporal dissimilarities within a sequence-identity cluster (e.g., Fig. 4, and Figs. 5C and 5D) can be highlighted automatically with the interactive Ananke UI, enabling detection of inconsistencies at a glance.

## CONCLUSIONS

Ananke is intended to complement, not replace, traditional sequence-identity-based approaches such as OTU clustering by examining the assumption that sequence similarity implies similar ecological properties. Using Ananke TSCs as a base, our work can be extended with deeper analyses of the relationships among Ananke TSCs. Future improvements to Ananke could include modifications of the distance measure or transformations of the time-series data that increase clustering performance with stationary temporal profiles and longer time series. The application of multivariate time-series analysis tools to Ananke TSCs will help quantify the importance of a TSC within the whole-community context and provide additional insight.

Ananke employs time-series clustering and interactive data exploration to highlight ecological events that can be obscured by alternative methods. We have demonstrated that Ananke can generate clusters of sequences that reflect ecological events such as enteric disease onset in the gut and seasonal changes in a lake. Ananke can also identify subtler patterns that would not be evident in taxonomic analyses, like the replacement of one strain by another of the same group (e.g., Fig. 3) or discordant dynamics among sequences of a single OTU (e.g., Fig. 4). Ananke represents a novel approach to analyzing longitudinal marker gene data with an emphasis on ecological relevance.

## ACKNOWLEDGEMENTS

We would like to thank the research group of Eric Alm for the generation of the human-associated data set we used in our validation trials of Ananke. We thank Fiona J. Whelan for testing the Ananke software and providing valuable input and bug reports. We thank the Earth Microbiome Project for sequencing the samples from Lake Mendota.

### Funding

MWH received support from the Natural Sciences and Engineering Research Council of Canada (NSERC). RGB received support from the Canada Research Chairs program and the NSERC Discovery Grants program. KDM received support from the United States National Science Foundation (NSF) Microbial Observatories program (MCB-0702395), the Long Term Ecological Research program (NTL-LTER DEB-1440297), an INSPIRE award (DEB-1344254), and the National Institute of Food and Agriculture, US Department of Agriculture (Hatch Project 1002996). The funders had no role in study design, data collection and analysis, decision to publish, or preparation of the manuscript.

### Grant Disclosures

The following grant information was disclosed by the authors:
Natural Sciences and Engineering Research Council of Canada (NSERC).
Canada Research Chairs program.
NSERC Discovery Grants program.
United States National Science Foundation (NSF) Microbial Observatories program: MCB-0702395.
Long Term Ecological Research program: NTL-LTER DEB-1440297.
INSPIRE award: DEB-1344254.
National Institute of Food and Agriculture.
US Department of Agriculture: Hatch Project: 1002996.

### Competing Interests

The authors declare there are no competing interests.

## Author Contributions

- Michael W. Hall conceived and designed the experiments, performed the experiments, analyzed the data, wrote the paper, prepared figures and/or tables, reviewed drafts of the paper.
- Robin R. Rohwer analyzed the data, wrote the paper, prepared figures and/or tables, reviewed drafts of the paper.
- Jonathan Perrie and Robert G. Beiko conceived and designed the experiments, performed the experiments, analyzed the data, wrote the paper, reviewed drafts of the paper.
- Katherine D. McMahon analyzed the data, wrote the paper, reviewed drafts of the paper.

## Data Availability

Hall, Michael; Rohwer, Robin; Perrie, Jonathan; McMahon, Katherine; Beiko, Robert (2017): Ananke Data Sets. figshare.

https://doi.org/10.6084/m9.figshare.c.3707938.v1.

## Supplemental Information

Supplemental information for this article can be found online at http://dx.doi.org/10.7717/peerj.3812#supplemental-information.

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
