# Peer review of "Ananke: temporal clustering reveals ecological dynamics of microbial communities"

_PeerJ, doi:10.7717/peerj.3812_

## Round 0.1 · original submission · Major Revisions

Dear Authors,

First, I would like to thank very much to the reviewers for dedicating their invaluable time to evaluate your work.

As you will see, the reviews contain multiple very important concerns, however, none of them suggest issues that are impossible to address. There *is* a growing appreciation for the importance of higher resolution analyses on marker gene amplicons, and I am sure your work will contribute to the evolution of the field. I hope you would consider submitting a revised manuscript.

The anonymous Reviewer #1 raises some ecologically important questions among other points such as additional documentation, inclusion of example data for test runs, changes in the text for clarity, and improvements in the introduction. Dr. Ramette (Reviewer #3) also raises multiple concerns regarding the methodology, and suggests improvements. Please make sure you address Dr. Ramette's questions regarding the appropriateness of the methods underlie your approach thoroughly.

We are looking forward to reading your revised work.


Best wishes,

Reviewer 1 ·

Basic reporting

The paper was well written and the figures were useful. No problems with basic reporting, but see comment about Preheim et al. 2013 in the general comments below which I think should be cited.

Experimental design

The research identifies a problem, and definitely addresses it. They perform some analyses based on simulated data that is nice and well described. Some of the short comings of the algorithm and future directions are clearly stated.

Validity of the findings

The conclusions are one of the strengths of the paper where they do a great job of recapping the positives and negatives of the approach.

Additional comments

Hall et al. Ananke Review

The contribution by Hall et al. provides an informative description of the bioinformatic approach they have developed. This software enables clustering of exact sequences across time-series based on their ‘dynamic similarity’. The paper is very well written and the software is a timely complement to OTU based approaches. The approach may be even more useful to help analyze datasets partitioned with finer resolution such as DADA2 and Minimum Entropy Decomposition, as more investigators transition their analyses to these methods. In addition to Ananke, the Shiny-based visualization is very cool, useful, and is key to assessing Ananke’s ability to discriminate temporally similar sequences.

Below I highlight some aspects that I find should be addressed, roughly, in order of importance.
An important point, in general, is that the authors seemed to leave out what would seem to be an important reference to work that is similar. They need to cite and discuss a paper by Preheim et al. 2013, “Distribution-Based Clustering: Using Ecology To Refine the Operational Taxonomic Unit.” The Preheim et al. software uses both sequence similarity and distribution of their abundance across samples, including discriminating above 97%, similarity as Ananke does. The authors need to explicitly address the similarities and dissimilarities between these two approaches, which seem quite similar. Ananke is different in that it does not require sequence similarities, so it can find co-occurrence clusters between non-closely-related taxa. Which may or may not be considered an advantage.

Line 300: Ananke does not perform consistently across datasets of variable lengths, and they speculate that it might differ based on sequencing depth and diversity. They suggest that the user needs to go use the visualization tools provided (which is very nice and works great) and set their cutoff thresholds based on detailed assessment of the data. However, I find this requirement at odds with their statement in the title of section at Line 103: “Unsupervised clustering of time-series distances”. I recommend omitting “Unsupervised”, here because even if they calculation is unsupervised (like OTU clustering), the assessment and analysis, does not seem to be so.

Section starting at Line 178: Title seems to disagree with my understanding of Line 95 and others. The authors found that more data meant less similarity to the “ground truth” not that data sets were “accurately clustered”. Authors state that the algorithm has a difficult time on taxa that are normally distributed with high or low variance, and on datasets with many time-frames, I appreciate their forthrightness. Though I recommend that the authors reflect this in the title of the section: Perhaps, “Assessing accuracy of Ananke with simulated datasets”

Line 319: I recommend the authors expand the point to include it’s ability to complement non-OTU based approaches like DADA2 (Callahan et al. ), MED (Eren et al.), and the Tikhonov method.

Line 214: Ananke cannot be used on taxa that are detected in less than 15 or 20% of samples. Given the importance of the rare biosphere, of course this is a drawback. I think that the authors should state this, but can also point out that these taxa will be found in the “noise bin”? It is a good example of why Ananke should be considered as a complementary approach to other methods, as stated in the conclusions.

Line 250: Many of these analyses may be performed on more powerful computers than a standard desktop. How is Ananke expected to perform on these rare taxa if one doesn’t filter them out? I imagine Ananke will perform well on some of these periodic taxa if they become very abundant on a few days, and not so well if they are just always around the limit of detection (very stochastic).

General comment: I think that the software would be more popular, accessible, and more cited if the authors wrote some additional guidance in the github page to give users examples of how Ananke could be used in parallel with other extremely common pipelines (QIIME and mothur) or (where I think it will be most useful) very quickly up-and-coming methods (MED and DADA2). This wouldn’t need to be in depth, just some general guidance.

Also, the software was very easy to install with root privileges, great job! Might consider guidance for users without root privileges. Also might consider providing a test dataset for testing if installation was fully successful and working as expected.

Line 53: Probably want to cite earlier work, e.g., Eren et al. 2013.

Line 57: The Callahan reference is now published in Nature Methods, 2016.

Line 63: Is Ananke an acronym for anything?

Line 230-232: Authors should keep in mind that some taxa may have multiple copies of 16S and sometimes the sequences can vary (I’m not sure about E. coli). This could be a type of positive control for their, or other investigators, analyses.

Section beginning at 298: The influence of the user’s selection of epsilon values on the results is very clear and appreciated.

Line 311 Aren’t Actinomycetales and Acidimicrobiales relatively closely related? Perhaps an indication of the sequence similarity of these taxa would be useful.

Reviewer 2 ·

Basic reporting

see General comments for the author

Experimental design

see General comments for the author

Validity of the findings

see General comments for the author

Additional comments

Hall et al. developed a software Ananke to cluster the time-series microbiome data. There are several major issues in this study.

The processing pipeline in Ananke consists of: 1) Tabulate unique sequences 2) Filter out sequences with low counts 3) Calculate short time-series distance and 4) Cluster using DBSCAN.

Unique sequences in the microbiome data can be easily calculated using 100% identity OTUs in the software QIIME or Mothur. Calculation of unique sequences was not performed in the previous studies, which was not due to the technical or software problems.

The aim of the operational taxonomic units (OTUs) clustering is not only the data simplification but also minimizing the influence of sequencing errors. The sequencing errors occur in all the sequencing platforms, particularly higher in 454 pyrosequencing. Using unique sequences rather than OTUs would make the analysis intolerance of sequencing errors. The 97% sequence similarity as OTU clustering cutoff was widely used in the 16S rRNA gene analysis, largely representing the community of the gut microbiome at the species level. 98% and 99% OTUs were used in the oral and skin microbiome studies as well. Unique sequences or 100% OTUs do not necessarily increase the resolution of the data analysis in the hypervariable regions of the 16S rRNA gene analysis.

In addition, Ananke takes the fasta file as input, ignoring the quality information (scores) of the sequencing data.

As shown on the simulated time-series data sets in the study, Ananke performed well with as few as ten time points. However, the data noise or error significantly affects Ananke over a large number of time points. Ananke has also the trouble in clustering normally distributed time-series data.

The analysis based on unique sequences significantly increased the computational memory requirement and the running time. The software performance, RAM usage and running time, was not evaluated in the study.

Thus all the above suggest that the rationale of the 1st step in the method does not make sense. It’s not a good idea to analyze the unique sequences in the microbiome study.

To make the method feasible, in the 2nd step, Ananke minimized the number of unique sequences by filtering out the ones with low abundance. A large proportion of data were missed in the downstream analysis, ignoring numerous species/strains in the microbiota and affecting the calculation of the microbial diversity. In contrast, the previous OTUs analysis utilized all the sequences in the samples. As shown in the study, a maximum of 157 and 635 Ananke TSCs were found with an average Ananke TSC comprising 0.6% and 0.2% of the data set in the human gut microbiome and in the freshwater lake microbiome, respectively. This result only represented a small subset of the microbial community. The clustering segregation is not apparent at all if TSCs are not manually highlighted in different colors.

Ananke uses the non-phylogenetic metrics (STS) in the 3rd step to compute the distances between unique sequences. However, the phylogenetic metrics, such as UniFrac distance, were more common in the previous microbiome studies. More metrics should be available in the software (http://qiime.org/1.6.0/scripts/beta_diversity_metrics.html).

The clustering parameter ε is hard to be chosen in 4th step of the software. The optimal number (or the robustness) of clusters can be estimated using the Calinski–Harabasz index and the silhouette analysis on k-means clustering. Ananke, as an unsupervised clustering though, needs to provide the measure to help the users determine the optimal number in a similar way.

Figure 1, Figure 3, Figure 4, Figure 5, Figure 6 and Figure S3 have no units and no exact scales labeled in all the y-axes and in some of x-axes. The results of the study cannot be evaluated.

The authors didn’t pairwise compare the results from Ananke to the ones identified in the original papers.

The paired-end Illumina sequencing has been widely used, however, Ananke cannot analyze this type of the data.

·

Basic reporting

no comment.

Experimental design

no comment.

Validity of the findings

1) The finding that 16S identity does not necessarily imply similar organismal ecology is not new, as the 16S rRNA gene has been shown since the discovery of its usefulness as a universal marker not to necessarily reflect the genomic diversity of the organisms (Abstract, Introduction). Yet, methods, such as Ananke, that aim at discovering finer ecological patterns are welcome.
2) The aim of the method is not clear: reading the Abstract and Introduction sections, it seems as if the authors propose to replace the current definition of OTU based on sequence similarity by clustering based on temporal profiles. Yet, the authors do not follow up in this direction in the Conclusion section (line 319), but seem to suggest otherwise its usefulness also for taxonomic purposes (e.g. lines 294-297).
3) The idea of OTU originates from the inability in microbiology to define the appropriate unit of evolution and selection. Therefore an "operational" definition is often used. Therefore one main question this study should discuss is whether a cluster of sequences defined based on a similar temporal pattern can constitute a unit of selection or evolution. How long should the time-series be then? How to standardize and compare across studies to compare those new temporally defined units? If we take several plant species for instance that are fluctuating similarly over time does this mean that they are under the same environmental triggers, under the same evolutionary pressure, etc.? I don’t think so, as this may be just a "guilt of association" case, especially if the time-series is short. Thus the important point of implications for evolution or selection should be discussed in the study. Also, the claim that “Ananke facilitates identification of potential microbial interactions" line 298 seems to go beyond what the tool can realistically provide (i.e. it can identify similar statistical patterns, not necessarily biological interactions). As a hypothesis generating/exploratory tool it seems however to be well suited.
Time was chosen as the main factor to cluster sequences, but one could envision the same being done using an environmental gradient instead. This approach is also not new and is what constitutes the core of the ecological modeling and community ecology disciplines.
4) Methodological aspects. The method computes short-term changes in slopes between sequence abundances at successive time points. Looking at the methodological procedure several points need to be particularly checked:
a. The sequence "abundances" are relative as they were standardized at each time point by sequence depth. Therefore, these abundances are not independent from each other anymore. The authors should consider including appropriate methods to deal with compositional data in their approach.
b. The equation line 97 seems not to be scaled, i.e. that if longer time-series are used the distances will de facto increase. In most computation of distances, there is a scaling factor to make sure that the distance is scaled to [0, 1] for instance to allow meaningful comparisons.
c. Lines 161-166, 214-245, 248-249: the procedure seems to arbitrarily exclude rare events or other types of sequences before computing further quantities. Therefore varying amount of singletons for instance which are then removed will lead to changes in relative abundances in the sequences that are kept for downstream calculations. Sequence depth is then affected.
d. The authors based their work on the Möller-Levet et al. 2003 paper that clearly states that the method they developed is meant to be used for short time-series with uneven sampling points. Yet, the authors of Ananke used long-term (191 time points, lines 212-213) and rather evenly sampled data (daily samples; line 134). Is the approach then appropriately applied?
e. In fact the authors used an OTU definition of 100% identity here because they investigated the changes in abundance profiles of unique sequences over time. What is the impact of organisms having several copies of the marker genes with potential sequence differences among the copies? (In the case of 16S rRNA operons, one would expect on average more than one copy per genome).
f. There is a danger in proposing to "explore individual patterns" (e.g. lines 325-326), when no overall test for temporal changes when all data is considered altogether is performed beforehand. This would be akin to performing multiple pairwise t-tests instead of initially determining whether the ANOVA for a whole dataset is significant. Using an appropriate omnibus test is the only way to reduce the chance to observe false positives just by chance. There are some techniques to deal with time series for whole communities (i.e. multivariate time-series) and the authors should probably compare their approach to those established methods.
Minor comments
-Line 120: "Normal distribution" should be better explained because it is not clear whether the authors want to model a constant mean and constant variance over time (stationarity) or changes in variance over time even though the mean is constant (non-stationarity). In figure 1, other simulated "patterns" describe changes over time (x axis), whereas “normal” seems to refer to what happens on the y axis.

---

## Round 0.2 · accepted · Accept

Thank you very much for addressing all reviewer concerns! Both reviewers are satisfied with your changes, and it is my pleasure to send your revised manuscript to the production team.

As a final note for post-publication efforts, I think it would have been beneficial for the user experience if Ananke was pip-installable. This would have required minimal additions to the current setup.py file, and I would be more than happy to help if you would like to pursue that and run into problems.

Best wishes,
Meren.

Reviewer 1 ·

Basic reporting

no comment

Experimental design

no comment

Validity of the findings

no comment

Additional comments

I appreciate the reviewers professional, complete, and satisfactory response to my initial review. The paper is well written, and Ananke is an excellent contribution. I commend them on the visualization code accompanying the manuscript that enables the user to easily plot and consider appropriate epsilon values for clustering.

·

Basic reporting

no comment

Experimental design

no comment

Validity of the findings

no comment

Additional comments

I do not have further comments and agree with the changes made by the authors.